# MULTI-AGENT CONSTRAINED POLICY OPTIMISATION

## ABSTRACT

Developing reinforcement learning algorithms that satisfy safety constraints is becoming increasingly important in real-world applications. In multi-agent reinforcement learning (MARL) settings, policy optimisation with safety awareness is particularly challenging because each individual agent has to not only meet its own safety constraints, but also consider those of others so that their joint behaviour can be guaranteed safe. Despite its importance, the problem of safe multi-agent learning has not been rigorously studied; very few solutions have been proposed, nor a sharable testing environment or benchmarks. To fill these gaps, in this work, we formulate the safe MARL problem as a constrained Markov game and solve it with policy optimisation methods. Our solutions—*Multi-Agent Constrained Policy Optimisation* (MACPO) and *MAPPO-Lagrangian*—leverage the theories from both constrained policy optimisation and multi-agent trust region learning. Crucially, our methods enjoy theoretical guarantees of both monotonic improvement in reward and satisfaction of safety constraints at every iteration. To examine the effectiveness of our methods, we develop the benchmark suite of *Safe Multi-Agent MuJoCo* that involves a variety of MARL baselines. Experimental results justify that MACPO/MAPPO-Lagrangian can consistently satisfy safety constraints, meanwhile achieving comparable performance to strong baselines.[1]

## 1 INTRODUCTION

In recent years, reinforcement learning (RL) techniques have achieved remarkable successes on a variety of complex tasks (Silver et al., 2016; 2017; Vinyals et al., 2019). Powered by deep neural networks, deep RL enables learning sophisticated behaviours. On the other hand, deploying neural networks turns the optimisation procedure from policy space to parameter space; this enables gradient-based methods to be applied (Sutton et al., 1999; Lillicrap et al., 2015; Schulman et al., 2017). For policy gradient methods, at every iteration, the parameters of a policy network are updated in the direction of the gradient that maximises return.

However, policies that are purely optimised for reward maximisation are rarely applicable to real-world problems. In many applications, an agent is often required not to visit certain states or take certain actions, which are thought of as "unsafe" either for itself or for other elements in the background (Moldovan & Abbeel, 2012; Achiam et al., 2017). For instance, a robot carrying materials in a warehouse should not damage its parts while delivering an item to a shelf, nor should a self-driving car cross on the red light while rushing towards its destination (Shalev-Shwartz et al., 2016). To tackle these issues, *Safe RL* (Moldovan & Abbeel, 2012; García & Fernández, 2015) is proposed, aiming to develop algorithms that learn policies that satisfy safety constraints. Despite the additional requirement of safety on solutions, algorithms with convergence guarantees have been proposed (Xu et al., 2021; Wei et al., 2021).

Developing safe policies for multi-agent systems is a challenging task. Part of the difficulty comes from solving multi-agent reinforcement learning (MARL) problems itself (Deng et al., 2021); more importantly, tackling safety in MARL is hard because each individual agent has to not only consider its own safety constraints, which already may conflict its reward maximisation, but also consider the safety constraints of others so that their joint behaviour is guaranteed to be safe. As a result, there are very few solutions that offer effective learning algorithms for safe MARL problems. In fact, many of the existing methods focus on learning to cooperate (Foerster et al., 2018; Rashid et al., 2018;

---

[1]Paper homepage including Videos and Code: `https://sites.google.com/view/macpo`

Yang & Wang, 2020). However, they often require certain structures on the solution; for example, Rashid et al. (2018) and Yang et al. (2020) adopt greedy maximisation on the local component of a monotonic joint value function, and Foerster et al. (2018) estimates the policy gradient based on the counterfactual value from a joint critic. Therefore, it is unclear how to directly incorporate safety constraints into these solution frameworks. Consequently, developing agents' collaborations towards reward maximisation under safety constraints remains an unsolved problem.

The goal of this paper is to increase practicality of MARL algorithms through endowing them with safety awareness. For this purpose, we introduce a general framework to formulate safe MARL problems, and solve them through multi-agent policy optimisation methods. Our solutions leverage techniques from both constrained policy optimisation (Achiam et al., 2017) and multi-agent trust region learning (Kuba et al., 2021a). The resulting algorithm attains properties of both monotonic improvement guarantee and constraints satisfaction guarantee at every iteration during training. To execute the optimisation objectives, we introduce two practical deep MARL algorithms: MACPO and MAPPO-Lagrangian. As a side contribution, we also develop the first safe MARL benchmark within the MuJoCo environment, which include a variety of MARL baseline algorithms. We evaluate MACPO and MAPPO-Lagrangian on a series of tasks, and results clearly confirm the effectiveness of our solutions both in terms of constraints satisfaction and reward maximisation. To our best knowledge, MACPO and MAPPO-Lagrangian are the first safety-aware model-free MARL algorithms that work effectively in the challenging MuJoCo tasks with safety constraints.

## 2 RELATED WORK

Considering safety in the development of AI is a long-standing topic (Amodei et al., 2016). When it comes to safe reinforcement learning (García & Fernández, 2015), a commonly used framework is *Constrained Markov Decision Processes* (CMDPs) (Altman, 1999). In a CMDP, at every step, in addition to the reward, the environment emits costs associated with certain constraints. As a result, the learning agent must try to satisfy those constraints while maximising the total reward. In general, the cost from the environment can be thought of as a measure of safety. Under the framework of CMDP, a safe policy is the one that explores the environment safely by keeping the total costs under certain thresholds. To tackle the learning problem in CMDPs, Achiam et al. (2017) introduced *Constrained Policy Optimisation* (CPO), which updates agent's policy under the trust region constraint (Schulman et al., 2015) to maximise surrogate return while obeying surrogate cost constraints. However, solving a constrained optimisation at every iteration of CPO can be cumbersome for implementation. An alternative solution is to apply primal-dual methods, giving rise to methods like TRPO-Lagrangian and PPO-Lagrangian (Ray et al., 2019). Although these methods achieve impressive performance in terms of safety, the performance in terms of reward is poor (Ray et al., 2019). Another class of algorithms that solves CMDPs is by Chow et al. (2018; 2019); these algorithms leverage the theoretical property of the Lyapunov functions and propose safe value iteration and policy gradient procedures. In contrast to CPO, Chow et al. (2018; 2019) can work with off-policy methods; they also can be trained end-to-end with no need for line search.

Safe multi-agent learning is an emerging research domain. Despite its importance (Shalev-Shwartz et al., 2016), there are few solutions that work with MARL in a model-free setting. The majority of methods are designed for robotics learning. For example, the technique of barrier certificates (Borrmann et al., 2015; Ames et al., 2016; Qin et al., 2020) or model predictive shielding (Zhang et al., 2019) from control theory is used to model safety. These methods, however, are specifically derived for robotics applications; they either are supervised learning based approaches, or require specific assumptions on the state space and environment dynamics. Moreover, due to the lack of a benchmark suite for safe MARL algorithms, the generalisation ability of those methods is unclear. The most related work to ours is Safe Dec-PG (Lu et al., 2021) where they used the primal-dual framework to find the saddle point between maximising reward and minimising cost. In particular, they proposed a decentralised policy descent-ascent method through a consensus network. However, reaching a consensus equivalently imposes an extra constraint of parameter sharing among neighbouring agents, which could yield suboptimal solutions (Kuba et al., 2021a). Furthermore, multi-agent policy gradient methods can suffer from high variance (Kuba et al., 2021b). In contrast, our methods employ trust region optimisation and do not assume any parameter sharing.

HATRPO (Kuba et al., 2021a) introduced the first multi-agent trust region method that enjoys theoretically-justified monotonic improvement guarantee. Its key idea is to make agents follow a

sequential policy update scheme so that the expected joint advantage will always be positive, thus increasing reward. In this work, we show how to further develop this theory and derive a protocol which, in addition to the monotonic improvement, also guarantees to satisfy the safety constraint at every iteration during learning. The resulting algorithm (Algorithm 1) successfully attains theoretical guarantees of both monotonic improvement in reward and satisfaction of safety constraints.

## 3  PROBLEM FORMULATION: CONSTRAINED MARKOV GAME

We formulate the safe MARL problem as a *constrained Markov game* $\langle \mathcal{N}, \mathcal{S}, \mathcal{A}, \mathrm{p}, \rho^0, \gamma, R, C, c \rangle$. Here, $\mathcal{N} = \{1, \ldots, n\}$ is the set of agents, $\mathcal{S}$ is the state space, $\mathcal{A} = \prod_{i=1}^n \mathcal{A}^i$ is the product of the agents' action spaces, known as the joint action space, $\mathrm{p} : \mathcal{S} \times \mathcal{A} \times \mathcal{S} \to \mathbb{R}$ is the probabilistic transition function, $\rho^0$ is the initial state distribution, $\gamma \in [0, 1)$ is the discount factor, $R : \mathcal{S} \times \mathcal{A} \to \mathbb{R}$ is the joint reward function, $C = \{C_j^i\}_{1 \le j \le m^i}^{i \in \mathcal{N}}$ is the set of sets of cost functions (every agent $i$ has $m^i$ cost functions) of the form $C_j^i : \mathcal{S} \times \mathcal{A}^i \to \mathbb{R}$, and finally the set of corresponding cost-constraining values is given by $c = \{c_j^i\}_{1 \le j \le m^i}^{i \in \mathcal{N}}$. At time step $t$, the agents are in a state $\mathrm{s}_t$, and every agent $i$ takes an action $\mathrm{a}_t^i$ according to its policy $\pi^i(\mathrm{a}^i | \mathrm{s}_t)$. Together with other agents' actions, it gives a joint action $\mathbf{a}_t = (\mathrm{a}_t^1, \ldots, \mathrm{a}_t^n)$ and the joint policy $\boldsymbol{\pi}(\mathbf{a} | \mathrm{s}) = \prod_{i=1}^n \pi^i(\mathrm{a}^i | \mathrm{s})$. The agents receive the reward $R(\mathrm{s}_t, \mathbf{a}_t)$, meanwhile each agent $i$ pays the costs $C_j^i(\mathrm{s}_t, \mathrm{a}_t^i), \forall j = 1, \ldots, m^i$. The environment then transits to a new state $\mathrm{s}_{t+1} \sim \mathrm{p}(\cdot | \mathrm{s}_t, \mathbf{a}_t)$. In this paper, we consider a *fully-cooperative* setting where all agents share the same reward function, aiming to maximise the expected total reward of

$$J(\boldsymbol{\pi}) \triangleq \mathbb{E}_{\mathrm{s}_0 \sim \rho^0, \mathbf{a}_{0:\infty} \sim \boldsymbol{\pi}, \mathrm{s}_{1:\infty} \sim \mathrm{p}} \Big[ \sum_{t=0}^{\infty} \gamma^t R(\mathrm{s}_t, \mathbf{a}_t) \Big],$$

meanwhile trying to satisfy every agent $i$'s safety constraints, written as

$$J_j^i(\boldsymbol{\pi}) \triangleq \mathbb{E}_{\mathrm{s}_0 \sim \rho^0, \mathbf{a}_{0:\infty} \sim \boldsymbol{\pi}, \mathrm{s}_{1:\infty} \sim \mathrm{p}} \Big[ \sum_{t=0}^{\infty} \gamma^t C_j^i(\mathrm{s}_t, \mathrm{a}_t^i) \Big] \le c_j^i, \quad \forall j = 1, \ldots, m^i. \tag{1}$$

We define the state-action value and the state-value functions in terms of reward as

$$Q_{\boldsymbol{\pi}}(s, \boldsymbol{a}) \triangleq \mathbb{E}_{\mathrm{s}_{1:\infty} \sim \mathrm{p}, \mathbf{a}_{1:\infty} \sim \boldsymbol{\pi}} \Big[ \sum_{t=0}^{\infty} \gamma^t R(\mathrm{s}_t, \mathbf{a}_t) \big| \mathrm{s}_0 = s, \mathbf{a}_0 = \boldsymbol{a} \Big], \quad \text{and} \quad V_{\boldsymbol{\pi}}(s) \triangleq \mathbb{E}_{\mathbf{a} \sim \boldsymbol{\pi}} \big[ Q_{\boldsymbol{\pi}}(s, \mathbf{a}) \big].$$

The joint policies $\boldsymbol{\pi}$ that satisfy the Inequality (1) are referred to as **feasible**. Notably, in the above formulation, although the action $\mathrm{a}_t^i$ of agent $i$ does not directly influence the costs $\{C_j^k(\mathrm{s}_t, \mathrm{a}_t^k)\}_{j=1}^{m^k}$ of other agents $k \ne i$, the action $\mathrm{a}_t^i$ will implicitly influence their total costs due to the dependence on the next state $\mathrm{s}_{t+1}$ [2]. For the $j^{\text{th}}$ cost function of agent $i$, we define the $j^{\text{th}}$ state-action cost value function and the state cost value function as

$$Q_{j,\boldsymbol{\pi}}^i(s, a^i) \triangleq \mathbb{E}_{\mathbf{a}^{-i} \sim \boldsymbol{\pi}^{-i}, \mathrm{s}_{1:\infty} \sim \mathrm{p}, \mathbf{a}_{1:\infty} \sim \boldsymbol{\pi}} \Big[ \sum_{t=0}^{\infty} \gamma^t C_j^i(\mathrm{s}_t, \mathrm{a}_t^i) \big| \mathrm{s}_0 = s, \mathrm{a}_0^i = a^i \Big],$$

$$\text{and,} \quad V_{j,\boldsymbol{\pi}}^i(s) \triangleq \mathbb{E}_{\mathbf{a} \sim \boldsymbol{\pi}, \mathrm{s}_{1:\infty} \sim \mathrm{p}, \mathbf{a}_{1:\infty} \sim \boldsymbol{\pi}} \Big[ \sum_{t=0}^{\infty} \gamma^t C_j^i(\mathrm{s}_t, \mathrm{a}_t^i) \big| \mathrm{s}_0 = s \Big], \quad \text{respectively.}$$

Notably, the cost value functions $Q_{j,\boldsymbol{\pi}}^i$ and $V_{j,\boldsymbol{\pi}}^i$, although similar to traditional $Q_{\boldsymbol{\pi}}$ and $V_{\boldsymbol{\pi}}$, involve extra indices $i$ and $j$; the superscript $i$ denotes an agent, and the subscript $j$ denotes its $j^{\text{th}}$ cost.

Throughout this work, we pay a close attention to the contribution to performance from different subsets of agents, therefore, we introduce the following notations. We denote an arbitrary subset

---

[2]We believe that this formulation realistically describes multi-agent interactions in the real-world; an action of an agent has an instantaneous effect on the system only locally, but the rest of agents may suffer from its consequences at later stages. For example, consider a car that crosses on the red light, although other cars may not be at risk of riding into pedestrians immediately, the induced traffic may cause hazards soon later.

$\{i_1, \ldots, i_h\}$ of agents as $i_{1:h}$; we write $-i_{1:h}$ to refer to its complement. Given the agent subset $i_{1:h}$, we define the *multi-agent state-action value function*:

$$Q_{\boldsymbol{\pi}}^{i_{1:h}}(s, \boldsymbol{a}^{i_{1:h}}) \triangleq \mathbb{E}_{\mathbf{a}^{-i_{1:h}} \sim \boldsymbol{\pi}^{-i_{1:h}}} \left[ Q_{\boldsymbol{\pi}}(s, \boldsymbol{a}^{i_{1:h}}, \mathbf{a}^{-i_{1:h}}) \right].$$

On top of it, the *multi-agent advantage function* [3] is defined as follows,

$$A_{\boldsymbol{\pi}}^{i_{1:h}}\left(s, \boldsymbol{a}^{j_{1:k}}, \boldsymbol{a}^{i_{1:h}}\right) \triangleq Q_{\boldsymbol{\pi}}^{j_{1:k}, i_{1:h}}\left(s, \boldsymbol{a}^{j_{1:k}}, \boldsymbol{a}^{i_{1:h}}\right) - Q_{\boldsymbol{\pi}}^{j_{1:k}}\left(s, \boldsymbol{a}^{j_{1:k}}\right). \tag{2}$$

An interesting fact about the above multi-agent advantage function is that any advantage $A_{\boldsymbol{\pi}}^{i_{1:h}}$ can be written as a sum of sequentially-unfolding multi-agent advantages of individual agents, that is,

**Lemma 1** (Multi-Agent Advantage Decomposition, Kuba et al. (2021b)). *For any state $s \in \mathcal{S}$, subset of agents $i_{1:h} \subseteq \mathcal{N}$, and joint action $\boldsymbol{a}^{i_{1:h}}$, the following identity holds*

$$A_{\boldsymbol{\pi}}^{i_{1:h}}\left(s, \boldsymbol{a}^{i_{1:h}}\right) = \sum_{j=1}^{h} A_{\boldsymbol{\pi}}^{i_j}\left(s, \boldsymbol{a}^{i_{1:j-1}}, a^{i_j}\right).$$

# 4 MULTI-AGENT CONSTRAINED POLICY OPTIMISATION

In this section, we first present a theoretically-justified safe multi-agent policy iteration procedure, which leverages multi-agent trust region learning and constrained policy optimisation to solve constrained Markov games. Based on this, we propose two practical deep MARL algorithms, enabling optimising neural-network based policies that satisfy safety constraints. Throughout this work, we refer the symbols $\boldsymbol{\pi}$ and $\bar{\boldsymbol{\pi}}$ to be the "current" and the "new" joint policies, respectively.

## 4.1 MULTI-AGENT TRUST REGION LEARNING WITH CONSTRAINTS

Kuba et al. (2021a) introduced the first multi-agent trust region method—HATRPO—that enjoys theoretically-justified monotonic improvement guarantee. Specifically, it relies on the multi-agent advantage decomposition in Lemma 1, and the "surrogate" return that is given as follows.

**Definition 1.** *Let $\boldsymbol{\pi}$ be a joint policy, $\bar{\boldsymbol{\pi}}^{i_{1:h-1}}$ be some other joint policy of agents $i_{1:h-1}$, and $\hat{\pi}^{i_h}$ be a policy of agent $i_h$. Then we define*

$$L_{\boldsymbol{\pi}}^{i_{1:h}}\left(\bar{\boldsymbol{\pi}}^{i_{1:h-1}}, \hat{\pi}^{i_h}\right) \triangleq \mathbb{E}_{s \sim \rho_{\boldsymbol{\pi}}, \mathbf{a}^{i_{1:h-1}} \sim \bar{\boldsymbol{\pi}}^{i_{1:h-1}}, a^{i_h} \sim \hat{\pi}^{i_h}} \left[ A_{\boldsymbol{\pi}}^{i_h}\left(s, \mathbf{a}^{i_{1:h-1}}, a^{i_h}\right) \right].$$

With the above definition, we can see that Lemma 1 allows for decomposing the joint surrogate return $L_{\boldsymbol{\pi}}(\bar{\boldsymbol{\pi}}) \triangleq \mathbb{E}_{s \sim \rho_{\boldsymbol{\pi}}, \mathbf{a} \sim \bar{\boldsymbol{\pi}}}[A_{\boldsymbol{\pi}}(s, \mathbf{a})]$ into a sum over surrogates of $L_{\boldsymbol{\pi}}^{i_{1:h}}(\bar{\boldsymbol{\pi}}^{i_{1:h-1}}, \bar{\pi}^{i_h})$, for $h = 1, \ldots, n$. This can be used to justify that if agents, with a joint policy $\boldsymbol{\pi}$, update their policies by following a *sequential update scheme*, that is, if each agent in the subset $i_{1:h}$ sequentially solves the following optimisation problem:

$$\bar{\pi}^{i_h} = \max_{\hat{\pi}^{i_h}} L_{\boldsymbol{\pi}}^{i_{1:h}}\left(\bar{\boldsymbol{\pi}}^{i_{1:h-1}}, \hat{\pi}^{i_h}\right) - \nu D_{\mathrm{KL}}^{\max}\left(\pi^{i_h}, \hat{\pi}^{i_h}\right),$$

$$\text{where} \quad \nu = \frac{4\gamma \max_{s,\boldsymbol{a}} |A_{\boldsymbol{\pi}}(s, \boldsymbol{a})|}{(1 - \gamma)^2}, \quad \text{and} \quad D_{\mathrm{KL}}^{\max}(\pi^{i_h}, \hat{\pi}^{i_h}) \triangleq \max_{s} D_{\mathrm{KL}}(\pi^{i_h}(\cdot|s), \hat{\pi}^{i_h}(\cdot|s)),$$

then the resulting joint policy $\bar{\boldsymbol{\pi}}$ will surely improve the expected return, i.e., $J(\bar{\boldsymbol{\pi}}) \geq J(\boldsymbol{\pi})$ (see the proof in Kuba et al. (2021a, Lemma 2)). We know that due to the penalty term $D_{\mathrm{KL}}^{\max}(\pi^{i_h}, \hat{\pi}^{i_h})$, the new policy $\bar{\pi}^{i_h}$ will stay close (*w.r.t* max-KL distance) to $\pi^{i_h}$.

For the safety constraints, we can extend Definition 1 to incorporate the "surrogate" cost, thus allowing us to study the cost functions in addition to the return.

**Definition 2.** *Let $\boldsymbol{\pi}$ be a joint policy, and $\bar{\pi}^i$ be some other policy of agent $i$. Then, for any of its costs of index $j \in \{1, \ldots, m^i\}$, we define*

$$L_{j, \boldsymbol{\pi}}^i\left(\bar{\pi}^i\right) = \mathbb{E}_{s \sim \rho_{\boldsymbol{\pi}}, a^i \sim \bar{\pi}^i} \left[ A_{j, \boldsymbol{\pi}}^i\left(s, a^i\right) \right].$$

---

[3]We would like to highlight that these multi-agent functions of $Q_{\boldsymbol{\pi}}^{i_{1:h}}$ and $A_{\boldsymbol{\pi}}^{i_{1:h}}$, although involve agents in superscripts, describe values *w.r.t* the reward rather than costs since they do not involve cost subscripts.

By generalising the result about the surrogate return in Equation (1), we can derive how the expected costs change when the agents update their policies. Specifically, we provide the following lemma.

**Lemma 2.** *Let $\pi$ and $\bar{\pi}$ be joint policies. Let $i \in \mathcal{N}$ be an agent, and $j \in \{1, \ldots, m^i\}$ be an index of one of its costs. The following inequality holds*

$$J_j^i(\bar{\pi}) \leq J_j^i(\pi) + L_{j,\pi}^i(\bar{\pi}^i) + \nu_j^i \sum_{h=1}^n D_{KL}^{max}(\pi^h, \bar{\pi}^h), \quad \text{where } \nu_j^i = \frac{4\gamma \max_{s,a^i} |A_{j,\pi}^i(s, a^i)|}{(1-\gamma)^2}.$$

See proof in Appendix A. The above lemma suggests that, as long as the distances between the policies $\pi^h$ and $\bar{\pi}^h$, $\forall h \in \mathcal{N}$, are sufficiently small, then the change in the $j^{\text{th}}$ cost of agent $i$, i.e., $J_j^i(\bar{\pi}) - J_j^i(\pi)$, is controlled by the surrogate $L_{j,\pi}^i(\bar{\pi}^i)$. Importantly, this surrogate is independent of other agents' new policies. Hence, when the changes in policies of all agents are sufficiently small, each agent $i$ can learn a better policy $\bar{\pi}^i$ by only considering its own surrogate return and surrogate costs. To summarise, we provide the following algorithm that guarantees both safety constraints satisfaction and monotonic performance improvement.

---

**Algorithm 1:** Safe Multi-Agent Policy Iteration with Monotonic Improvement Property

1: Initialise a safe joint policy $\pi_0 = (\pi_0^1, \ldots, \pi_0^n)$.
2: **for** $k = 0, 1, \ldots$ **do**
3:     Compute the advantage functions $A_{\pi_k}(s, \boldsymbol{a})$ and $A_{j,\pi_k}^i(s, a^i)$, for all state-(joint)action pairs $(s, \boldsymbol{a})$, agents $i$, and constraints $j \in \{1, \ldots, m^i\}$.
4:     Compute $\nu = \frac{4\gamma \max_{s,\boldsymbol{a}} |A_{\pi_k}(s,\boldsymbol{a})|}{(1-\gamma)^2}$, and $\nu_j^i = \frac{4\gamma \max_{s,a^i} |A_{j,\pi_k}^i(s,a^i)|}{(1-\gamma)^2}$, $\forall i \in \mathcal{N}, j = 1, \ldots, m^i$.
5:     Draw a permutaion $i_{1:n}$ of agents at random.
6:     **for** $h = 1 : n$ **do**
7:         Compute the radius of the KL-constraint $\delta^{i_h}$  // *see Appendix B for the setup of $\delta^{i_h}$.*
8:         Make an update $\pi_{k+1}^{i_h} = \arg\max_{\pi^{i_h} \in \overline{\Pi}^{i_h}} \left[ L_{\pi_k}^{i_{1:h}}\left(\pi_{k+1}^{i_{1:h-1}}, \pi^{i_h}\right) - \nu D_{KL}^{max}\left(\pi_k^{i_h}, \pi^{i_h}\right) \right]$,

    where $\overline{\Pi}^{i_h}$ is a subset of safe policies of agent $i_h$, given by

$$\overline{\Pi}^{i_h} = \left\{ \pi^{i_h} \in \Pi^{i_h} \mid D_{KL}^{max}(\pi_k^{i_h}, \pi^{i_h}) \leq \delta^{i_h}, \text{ and} \right.$$

$$\left. J_j^{i_h}(\pi_k) + L_{j,\pi_k}^{i_h}(\pi^{i_h}) + \nu_j^{i_h} D_{KL}^{max}(\pi_k^{i_h}, \pi^{i_h}) \leq c_j^{i_h} - \sum_{l=1}^{h-1} \nu_j^{i_l} D_{KL}^{max}(\pi_k^{i_l}, \pi^{i_l}), \forall j = 1, \ldots, m^{i_h} \right\}.$$

9:     **end for**
10: **end for**

---

In the above algorithm, in addition to sequentially maximising agents' surrogate returns, the agents must assure that their surrogate costs stay below the corresponding safety thresholds. Meanwhile, they have to constrain their policy search to small local neighbourhoods (*w.r.t* max-KL distance). As such, Algorithm 1 demonstrates two desirable properties: reward performance improvement and satisfaction of safety constraints, which we justify in the following theorem.

**Theorem 1.** *If a sequence of joint policies $(\pi_k)_{k=0}^{\infty}$ is obtained from Algorithm 1, then it has the monotonic improvement property, $J(\pi_{k+1}) \geq J(\pi_k)$, as well as it satisfies the safety constraints, $J_j^i(\pi_k) \leq c_j^i$, for all $k \in \mathbb{N}, i \in \mathcal{N}$, and $j \in \{1, \ldots, m^i\}$.*

See proof in Appendix B. The above theorem assures that agents that follow Algorithm 1 will only explore safe policies; meanwhile, every new policy will be guaranteed to result in performance improvement. These two properties hold under the conditions that only restrictive policy updates are made; this is due to the KL-penalty term in every agent's objective (i.e., $\nu D_{KL}^{max}(\pi_k^{i_h}, \pi^{i_h})$), as well as the constraints on cost surrogates (i.e., the conditions in $\overline{\Pi}^{i_h}$). In practice, it can be intractable to evaluate $D_{KL}\left(\pi_k^{i_h}(\cdot|s), \pi^{i_h}(\cdot|s)\right)$ at every state in order to compute $D_{KL}^{max}(\pi_k^{i_h}, \pi^{i_h})$. In the following subsections, we describe how we can approximate Algorithm 1 in the case of parameterised policies, similar to TRPO/PPO implementations (Schulman et al., 2015; 2017).

### 4.2 MACPO: Multi-Agent Constrained Policy Optimisation

Here we focus on the practical settings where large state and action spaces prevent agents from designating policies $\pi^i(\cdot|s)$ for each state separately. To handle this, we parameterise each agent's $\pi^i_{\theta^i}$ by a neural network $\theta^i$. Correspondingly, the joint policies $\pi_\theta$ are parametrised by $\theta = (\theta^1, \ldots, \theta^n)$.

Let's recall that at every iteration of Algorithm 1, every agent $i_h$ maximises its surrogate return with a KL-penalty, subject to surrogate cost constraint. Yet, direct computation of the max-KL constraint is intractable in practical settings, as it would require computation of KL-divergence at every single state. Instead, one can relax it by adopting a form of expected KL-constraint $\overline{D}_{\mathrm{KL}}(\pi^{i_h}_k, \pi^{i_h}) \leq \delta$ where $\overline{D}_{\mathrm{KL}}(\pi^{i_h}_k, \pi^{i_h}) \triangleq \mathbb{E}_{s \sim \rho_{\pi_k}} \left[ D_{\mathrm{KL}}(\pi^{i_h}_k(\cdot|s), \pi^{i_h}(\cdot|s)) \right]$. Such an expectation can be approximated by stochastic sampling. As a result, the optimisation problem solved by agent $i_h$ is written as

$$\theta^{i_h}_{k+1} = \arg \max_{\theta^{i_h}} \mathbb{E}_{s \sim \rho_{\pi_{\theta_k}}, \mathbf{a}^{i_{1:h-1}} \sim \pi^{i_{1:h-1}}_{\theta^{i_{1:h-1}}_{k+1}}, a^{i_h} \sim \pi^{i_h}_{\theta^{i_h}}} \left[ A^{i_h}_{\pi_{\theta_k}} \left(s, \mathbf{a}^{i_{1:h-1}}, a^{i_h}\right) \right]$$

$$\text{s.t.} \quad J^{i_h}_j \left(\pi_{\theta_k}\right) + \mathbb{E}_{s \sim \rho_{\pi_{\theta_k}}, a^{i_h} \sim \pi^{i_h}_{\theta^{i_h}_k}} \left[ A^{i_h}_{j, \pi_{\theta_k}} \left(s, a^{i_h}\right) \right] \leq c^{i_h}_j, \; \forall j \in \{1, \ldots, m^{i_h}\},$$

$$\text{and} \quad \overline{D}_{\mathrm{KL}}\left(\pi^{i_h}_k, \pi^{i_h}\right) \leq \delta. \tag{3}$$

We can further approximate Equation (3) by Taylor expansion of the optimisation objective and cost constraints up to the first order, and the KL-divergence up to the second order. Consequently, the optimisation problem can be written as

$$\theta^{i_h}_{k+1} = \arg \max_{\theta^{i_h}} \left(g^{i_h}\right)^T \left(\theta^{i_h} - \theta^{i_h}_k\right)$$

$$\text{s.t.} \quad d^{i_h}_j + \left(b^{i_h}_j\right)^T \left(\theta^{i_h} - \theta^{i_h}_k\right) \leq 0, \quad j = 1, \ldots, m$$

$$\text{and} \quad \frac{1}{2} \left(\theta^{i_h} - \theta^{i_h}_k\right)^T H^{i_h} \left(\theta^{i_h} - \theta^{i_h}_k\right) \leq \delta, \tag{4}$$

where $g^{i_h}$ is the gradient of the objective of agent $i_h$ in Equation (3), $d^i_j = J^i_j(\pi_{\theta_k}) - c^i_j$, and $H^{i_h} = \nabla^2_{\theta^{i_h}} \overline{D}_{\mathrm{KL}}(\pi^{i_h}_{\theta^{i_h}_k}, \pi^{i_h})\big|_{\theta^{i_h} = \theta^{i_h}_k}$ is the Hessian of the average KL divergence of agent $i_h$, and $b^{i_h}_j$ is the gradient of agent of the $j^{\mathrm{th}}$ constraint of agent $i_h$.

Similar to Chow et al. (2017) and Achiam et al. (2017), one can take a primal-dual optimisation approach to solve the linear quadratic optimisation in Equation (4). Specifically, the dual form can be written as:

$$\max_{\lambda^{i_h} \geq 0, \mathbf{v}^{i_h} \geq 0} \frac{-1}{2\lambda^{i_h}} \left[ (g^{i_h})^T (H^{i_h})^{-1} g^{i_h} - 2(r^{i_h})^T \mathbf{v}^{i_h} + (\mathbf{v}^{i_h})^T S^{i_h} \right] + (\mathbf{v}^{i_h})^T c^{i_h} - \frac{\lambda^{i_h} \delta}{2},$$

$$\text{where} \quad r^{i_h} \triangleq (g^{i_h})^T (H^{i_h})^{-1} B^{i_h}, \; B^{i_h} = \left[ b^{i_h}_1, \ldots, b^{i_h}_m \right] \text{ and } S^{i_h} \triangleq (B^{i_h})^T (H^{i_h})^{-1} B^{i_h}. \tag{5}$$

Given the solution to the dual form in Equation (5), i.e., $\lambda^{i_h}_*$ and $\mathbf{v}^{i_h}_*$, the solution to the primal problem in Equation (4) can thus be written by

$$\theta^{i_h}_* = \theta^{i_h}_k + \frac{1}{\lambda^{i_h}_*} \left(H^{i_h}\right)^{-1} \left(g^{i_h} - B^{i_h} \mathbf{v}^{i_h}_*\right).$$

In practice, we use backtracking line search starting at $1/\lambda^{i_h}_*$ to choose the step size of the above update. Furthermore, we note that the optimisation step in Equation (4) is an approximation to the original problem from Equation (3); therefore, it is possible that an infeasible policy $\pi_{\theta^{i_h}_{k+1}}$ will be generated. Fortunately, as the policy optimisation takes place in the trust region of $\pi^{i_h}_{\theta^{i_h}_k}$, the size of update is small, and a feasible policy can be easily recovered. In particular, for problems with one safety constraint, i.e., $m^{i_h} = 1$, one can recover a feasible policy by applying a TRPO step on the cost surrogate, written as

$$\theta^{i_h}_{k+1} = \theta^{i_h}_k - \alpha^j \sqrt{\frac{2\delta}{b^{i_h T} (H^{i_h})^{-1} b^{i_h}}} \left(H^{i_h}\right)^{-1} b^{i_h} \tag{6}$$

where $\alpha^j$ is adjusted through backtracking line search. To put it together, we refer to this algorithm as *MACPO*, and provide its pseudocode in Appendix D.

## 4.3 MAPPO-Lagrangian

In addition to MACPO, one can use Lagrangian multipliers in place of optimisation with linear and quadratic constraints to solve Equation (3). The Lagrangian method is simple to implement, and it does not require computations of the Hessian $\boldsymbol{H}^{i_h}$ whose size grows quadratically with the dimension of the parameter vector $\theta^{i_h}$.

Before we proceed, let us briefly recall the optimisation procedure with a Lagrangian multiplier. Suppose that our goal is to maximise a bounded real-valued function $f(x)$ under a constraint $g(x)$; $\max_x f(x)$, s.t. $g(x) \leq 0$. We can introduce a scalar variable $\lambda$ and reformulate the optimisation by

$$\max_x \min_{\lambda \geq 0} f(x) - \lambda g(x). \tag{7}$$

Suppose that $x_+$ satisfies $g(x_+) > 0$. This immediately implies that $-\lambda g(x_+) \to -\infty$, as $\lambda \to +\infty$, and so Equation (7) equals $-\infty$ for $x = x_+$. On the other hand, if $x_-$ satisfies $g(x_-) \leq 0$, we have that $-\lambda g(x_-) \geq 0$, with equality only for $\lambda = 0$. In that case, the optimisation objective's value equals $f(x_-) > -\infty$. Hence, the only candidate solutions to the problem are those $x$ that satisfy the constraint $g(x) \leq 0$, and the objective matches with $f(x)$.

We can employ the above trick to the constrained optimisation problem from Equation (3) by subsuming the cost constraints into the optimisation objective with Lagrangian multipliers. As such, agent $i_h$ computes $\bar{\lambda}^{i_h}_{1:m^{i_h}}$ and $\theta^{i_h}_{k+1}$ to solve the following min-max optimisation problem

$$\min_{\lambda^{i_h}_{1:m^{i_h}} \geq 0} \max_{\theta^{i_h}} \left[ \mathbb{E}_{s \sim \rho_{\pi_{\theta_k}}, \mathbf{a}^{i_{1:h-1}} \sim \pi^{i_{1:h-1}}_{\theta^{i_{1:h-1}}_{k+1}}, a^{i_h} \sim \pi^{i_h}_{\theta^{i_h}}} \left[ A^{i_h}_{\pi_{\theta_k}} \left( s, \mathbf{a}^{i_{1:h-1}}, a^{i_h} \right) \right] \right.$$

$$\left. - \sum_{u=1}^{m^{i_h}} \lambda^{i_h}_u \left( \mathbb{E}_{s \sim \rho_{\pi_{\theta_k}}, a^{i_h} \sim \pi^{i_h}_{\theta^{i_h}}} \left[ A^{i_h}_{u, \pi_{\theta_k}} \left( s, a^{i_h} \right) \right] + d^{i_h}_u \right) \right],$$

$$\text{s.t. } \overline{D}_{\text{KL}} \left( \pi^{i_h}_{\theta^{i_h}_k}, \pi^{i_h}_{\theta^{i_h}} \right) \leq \delta. \tag{8}$$

Although the objective from Equation (8) is affine in the Lagrangian multipliers $\lambda^{i_h}_u$ $(u = 1, \ldots, m^{i_h})$, which enables gradient-based optimisation solutions, computing the KL-divergence constraint still complicates the overall process. To handle this, one can further simplify it by adopting the *PPO-clip* objective (Schulman et al., 2017), which enables replacing the KL-divergence constraint with the *clip* operator, and update the policy parameter with first-order methods. We do so by defining

$$A^{i_h,(\lambda)}_{\pi_{\theta_k}} \left( s, \boldsymbol{a}^{i_{1:h-1}}, a^{i_h} \right) \triangleq A^{i_h}_{\pi_{\theta_k}} \left( s, \boldsymbol{a}^{i_{1:h-1}}, a^{i_h} \right) - \sum_{u=1}^{m^{i_h}} \lambda^{i_h}_u \left( A^{i_h}_{u, \pi_{\theta_k}} \left( s, a^{i_h} \right) + d^{i_h}_u \right),$$

and rewriting the Equation (8) as

$$\min_{\lambda^{i_h}_{1:m^{i_h}} \geq 0} \max_{\theta^{i_h}} \mathbb{E}_{s \sim \rho_{\pi_{\theta_k}}, \boldsymbol{a}^{i_{1:h-1}} \sim \pi^{i_{1:h-1}}_{\theta^{i_{1:h-1}}_{k+1}}, a^{i_h} \sim \pi^{i_h}_{\theta^{i_h}}} \left[ A^{i_h,(\lambda)}_{\pi_{\theta_k}} \left( s, \mathbf{a}^{i_{1:h-1}}, a^{i_h} \right) \right],$$

$$\text{s.t. } \overline{D}_{\text{KL}} \left( \pi^{i_h}_{\theta^{i_h}_k}, \pi^{i_h}_{\theta^{i_h}} \right) \leq \delta. \tag{9}$$

The objective in Equation (9) takes a form of an expectation with quadratic constraint on the policy. Up to the error of approximation of KL-constraint with the *clip* operator, it can be equivalently transformed into an optimisation of a clipping objective. Finally, the objective takes the form of

$$\mathbb{E}_{s \sim \rho_{\pi_{\theta_k}}, \boldsymbol{a}^{i_{1:h-1}} \sim \pi^{i_{1:h-1}}_{\theta^{i_{1:h-1}}_{k+1}}, a^{i_h} \sim \pi^{i_h}_{\theta^{i_h}_k}} \left[ \min \left( \frac{\pi^{i_h}_{\theta^{i_h}} (a^{i_h} | s)}{\pi^{i_h}_{\theta^{i_h}_k} (a^{i_h} | s)} A^{i_h,(\lambda)}_{\pi_{\theta_k}} \left( s, \boldsymbol{a}^{i_{1:h-1}}, a^{i_h} \right), \right.\right.$$

$$\left.\left. \text{clip} \left( \frac{\pi^{i_h}_{\theta^{i_h}} (a^{i_h} | s)}{\pi^{i_h}_{\theta^{i_h}_k} (a^{i_h} | s)}, 1 \pm \epsilon \right) A^{i_h,(\lambda)}_{\pi_{\theta_k}} \left( s, \boldsymbol{a}^{i_{1:h-1}}, a^{i_h} \right) \right) \right]. \tag{10}$$

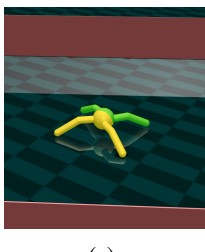 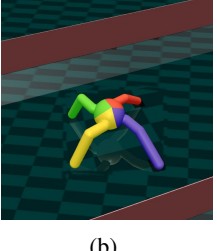 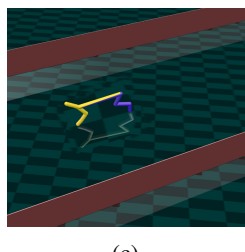

(a)             (b)             (c)

Figure 1: Example tasks in Safe Multi-Agent MuJoCo Environment. (a): Safe 2x4-Ant, (b): Safe 4x2-Ant, (c): Safe 2x3-HalfCheetah. Body parts of different colours are controlled by different agents. Agents jointly learn to manipulate the robot, while avoiding crashing into unsafe red areas.

The clip operator replaces the policy ratio with $1 - \epsilon$, or $1 + \epsilon$, depending on whether its value is below or above the threshold interval. As such, agent $i_h$ can learn within its trust region by updating $\theta^{i_h}$ to maximise Equation (10), while the Lagrangian multipliers are updated towards the direction opposite to their gradients of Equation (8), which can be computed analytically. We refer to this algorithm as *MAPPO-Lagrangian*, and give a detailed pseudocode of it in Appendix E.

## 5   EXPERIMENTS

Although MARL researchers have long had a variety of environments to test different algorithms, such as StarCraftII (Samvelyan et al., 2019) and Multi-Agent MuJoCo (Peng et al., 2020), no public safe MARL benchmark has been proposed; this impedes researchers from evaluating and benchmarking safety-aware multi-agent learning methods. As a key contribution of this paper, we introduce *Safe Multi-Agent MuJoCo Benchmark*, a safety-aware extension of the MuJoCo environment that is designed for safe MARL research. We show example tasks in Figure 1, in our environment, safety-aware agents have to learn not only skilful manipulations of a robot, but also to avoid crashing into unsafe obstacles and positions. For more details of the setup, please refer to Appendix F.

We use Safe MAMuJoCo to examine if the MACPO/MAPPO-Lagrangian agents can satisfy their safety constraints and cooperatively learn to achieve high rewards, compared to existing MARL algorithms. Notably, our proposed methods adopt two different approaches for achieving safety. MACPO reaches safety via *hard* constraints and backtracking line search, while MAPPO-Lagrangian maintains a rather *soft* safety awareness by performing gradient descents on the PPO-clip objective. Figure 2 shows cost and reward performance comparisons between MACPO, MAPPO-Lagrangian, MAPPO (Yu et al., 2021), IPPO (de Witt et al., 2020), and HAPPO (Kuba et al., 2021a) algorithms on three challenging tasks. Figure 2 should be interpreted at three-folds; each subfigure represents a different robot, within each subfigure, three task setups in terms of multi-agent control are considered, for each task, we plot the cost curves (the lower the better) in the upper row, and plot the reward curves (the higher the better) in the bottom row. Detailed hyperparameter settings are described in Appendix H.

The experiments reveal that both MACPO and MAPPO-Lagrangian quickly learn to satisfy safety constraints, and keep their explorations within the feasible policy space. This stands in contrast to IPPO, MAPPO, and HAPPO which largely violate the constraints thus being unsafe. Furthermore, our algorithms achieve comparable reward scores; both methods are often better than IPPO. In general, the performance (in terms of reward) of MAPPO-Lagrangian is better than of MACPO; moreover, MAPPO-Lagrangian outperforms the unconstrained MAPPO on challenging *Ant* tasks. We note that on none of the tasks the reward of HAPPO was exceeded though it is unsafe.

## 6   CONCLUSION

In this paper, we tackled multi-agent policy optimisation problems with safety constraints. Central to our findings is the safe multi-agent policy iteration procedure that attains theoretically-justified monotonic improvement guarantee and constraints satisfaction guarantee at every iteration during learning. Based on this, we proposed two practical algorithms: MACPO and MAPPO-Lagrangian. To demonstrate their effectiveness, we introduced a new benchmark suite of *Safe Multi-Agent MuJoCo* and compared our methods against strong MARL baselines. Results show that both of our methods can significantly outperform existing state-of-the-art methods such as IPPO, MAPPO and HAPPO in terms of safety, meanwhile maintaining comparable performance in terms of reward.

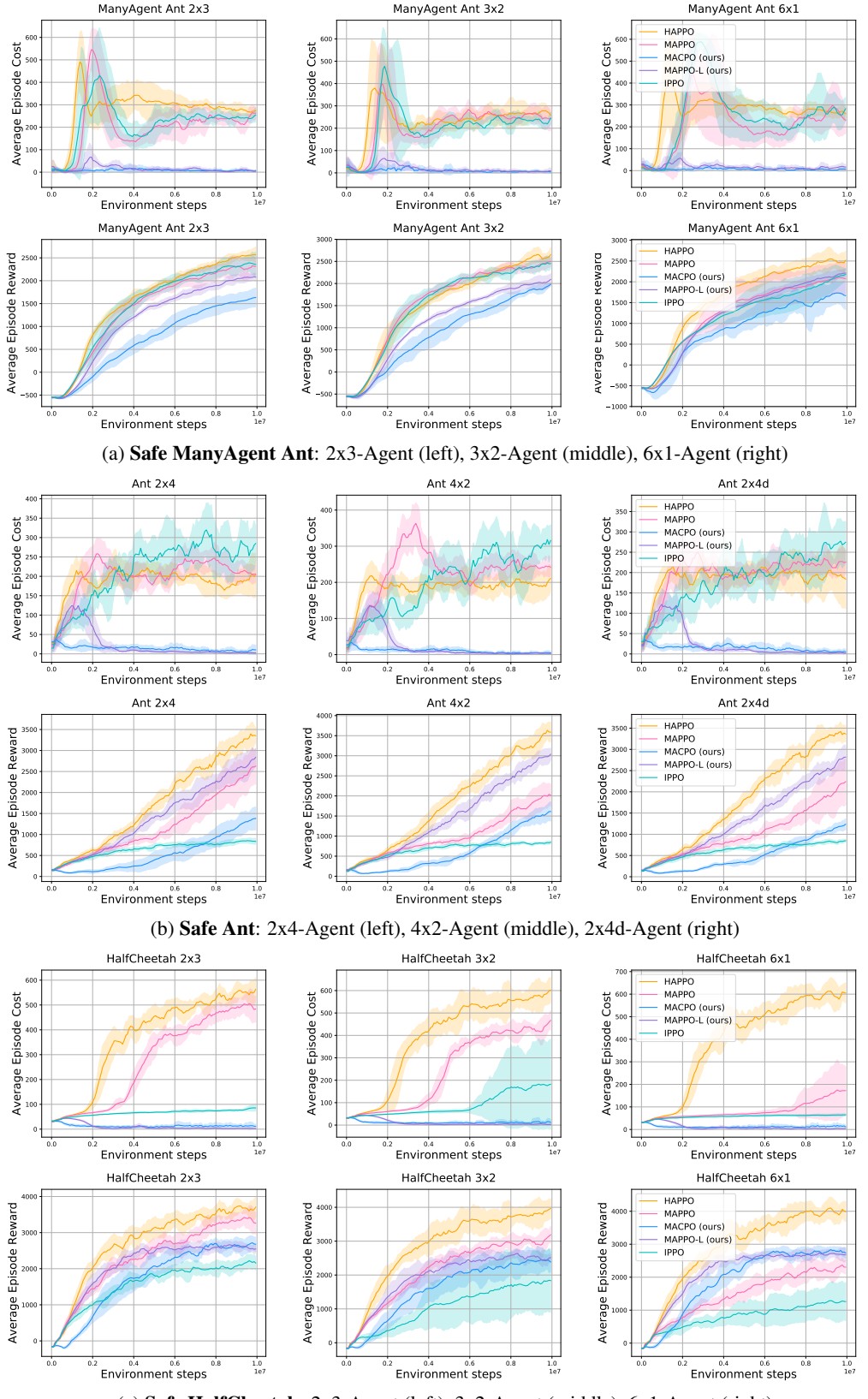

(a) **Safe ManyAgent Ant**: 2x3-Agent (left), 3x2-Agent (middle), 6x1-Agent (right)

(b) **Safe Ant**: 2x4-Agent (left), 4x2-Agent (middle), 2x4d-Agent (right)

(c) **Safe HalfCheetah**: 2x3-Agent (left), 3x2-Agent (middle), 6x1-Agent (right)

Figure 2: Performance comparisons on tasks of Safe ManyAgent Ant, Safe Ant, and Safe HalfChee-tah in terms of cost (the first row) and reward (the second row). The safety constraint values are: 1 for ManyAgent Ant, 0.2 for Ant, and 5 for HalfCheetah. Our methods consistently achieve almost zero costs, thus satisfying safe constraints, on all tasks. In terms of reward, our methods outperform IPPO and MAPPO on some tasks but underperform HAPPO, which is also an unsafe algorithm.

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

# Appendices

## A    Proofs of preliminary results

**Lemma 1** (Multi-Agent Advantage Decomposition, Kuba et al. (2021b)). *For any state $s \in \mathcal{S}$, subset of agents $i_{1:h} \subseteq \mathcal{N}$, and joint action $\boldsymbol{a}^{i_{1:h}}$, the following identity holds*

$$A_\pi^{i_{1:h}}\left(s, \boldsymbol{a}^{i_{1:h}}\right) = \sum_{j=1}^{h} A_\pi^{i_j}\left(s, \boldsymbol{a}^{i_{1:j-1}}, a^{i_j}\right).$$

*Proof.* We write the multi-agent advantage as in its definition, and expand it in a telescoping sum.

$$
\begin{aligned}
A_\pi^{i_{1:h}}\left(s, \boldsymbol{a}^{i_{1:h}}\right) &= Q_\pi^{i_{1:h}}\left(s, \boldsymbol{a}^{i_{1:h}}\right) - V_\pi(s) \\
&= \sum_{j=1}^{h}\left[Q_\pi^{i_{1:j}}\left(s, \boldsymbol{a}^{i_{1:j}}\right) - Q_\pi^{i_{1:j-1}}\left(s, \boldsymbol{a}^{i_{1:j-1}}\right)\right] \\
&= \sum_{j=1}^{h} A_\pi^{i_j}\left(s, \boldsymbol{a}^{i_{1:j-1}}, a^{i_j}\right).
\end{aligned}
$$

$\square$

**Lemma 2.** *Let $\pi$ and $\bar{\pi}$ be joint policies. Let $i \in \mathcal{N}$ be an agent, and $j \in \{1, \ldots, m^i\}$ be an index of one of its costs. The following inequality holds*

$$J_j^i(\bar{\pi}) \le J_j^i(\pi) + L_{j,\pi}^i\left(\bar{\pi}^i\right) + \nu_j^i \sum_{h=1}^{n} D_{KL}^{max}\left(\pi^h, \bar{\pi}^h\right), \quad \text{where } \nu_j^i = \frac{4\gamma \max_{s,a^i}\left|A_{j,\pi}^i(s, a^i)\right|}{(1-\gamma)^2}.$$

*Proof.* From the proof of Theorem 1 from Schulman et al. (2015) (in particular, equations (41)-(45)), applied to joint policies $\pi$ and $\bar{\pi}$, we conclude that

$$
\begin{aligned}
J_j^i(\bar{\pi}) &\le J_j^i(\pi) + \mathbb{E}_{s\sim\rho_\pi, \mathbf{a}\sim\bar{\pi}}\left[A_{j,\pi}^i(s, a^i)\right] + \frac{4\alpha^2\gamma \max_{s,a^i}\left|A_{j,\pi}^i(s, a^i)\right|}{(1-\gamma)^2}, \\
\text{where} \quad & \alpha = D_{\mathrm{TV}}^{\max}(\pi, \bar{\pi}) = \max_s D_{\mathrm{TV}}(\pi(\cdot|s), \bar{\pi}(\cdot|s)).
\end{aligned}
$$

Using the inequality $D_{\mathrm{TV}}(p, q)^2 \le D_{\mathrm{KL}}(p, q)$ (Pollard, 2000; Schulman et al., 2015), we obtain

$$
\begin{aligned}
J_j^i(\bar{\pi}) &\le J_j^i(\pi) + \mathbb{E}_{s\sim\rho_\pi, \mathbf{a}\sim\bar{\pi}}\left[A_{j,\pi}^i(s, a^i)\right] + \frac{4\gamma \max_{s,a^i}\left|A_{j,\pi}^i(s, a^i)\right|}{(1-\gamma)^2} D_{\mathrm{KL}}^{\max}(\pi, \bar{\pi}), \\
\text{where} \quad & D_{\mathrm{KL}}^{\max}(\pi, \bar{\pi}) = \max_s D_{\mathrm{KL}}(\pi(\cdot|s), \bar{\pi}(\cdot|s)).
\end{aligned}
$$

Notice now that we have $\mathbb{E}_{s\sim\rho_\pi, \mathbf{a}\sim\bar{\pi}}\left[A_{j,\pi}^i(s, a^i)\right] = \mathbb{E}_{s\sim\rho_\pi, a^i\sim\bar{\pi}^i}\left[A_{j,\pi}^i(s, a^i)\right]$, and

$$
\begin{aligned}
D_{\mathrm{KL}}^{\max}(\pi, \bar{\pi}) = \max_s D_{\mathrm{KL}}(\pi(\cdot|s), \bar{\pi}(\cdot|s)) &= \max_s\left(\sum_{l=1}^{n} D_{\mathrm{KL}}\left(\pi^l(\cdot|s), \bar{\pi}^l(\cdot|s)\right)\right) \\
&\le \sum_{l=1}^{n} \max_s D_{\mathrm{KL}}\left(\pi^l(\cdot|s), \bar{\pi}^l(\cdot|s)\right) = \sum_{l=1}^{n} D_{\mathrm{KL}}^{\max}\left(\pi^l, \bar{\pi}^l\right).
\end{aligned}
\tag{11}
$$

Setting $\nu_j^i = \frac{4\gamma \max_{s,a^i}\left|A_{j,\pi}^i(s, a^i)\right|}{(1-\gamma)^2}$, we finally obtain

$$J_j^i(\bar{\pi}) \le J_j^i(\pi) + L_{j,\pi}^i\left(\bar{\pi}^i\right) + \nu_j^i \sum_{l=1}^{n} D_{\mathrm{KL}}^{\max}\left(\pi^l, \bar{\pi}^l\right).$$

$\square$

## B    Auxiliary Results for Algorithm 1

**Remark 1.** *In Algorithm 1, we compute the size of KL constraint as*

$$
\delta^{i_h} = \min \left\{ \min_{l \leq h-1} \min_{1 \leq j \leq m^l} \frac{c_j^{i_l} - J_j^{i_l}(\boldsymbol{\pi}_k) - L_{j,\boldsymbol{\pi}_k}^{i_l}(\pi_{k+1}^{i_l}) - \nu_j^{i_l} \sum_{u=1}^{h-1} D_{KL}^{max}(\pi_k^u, \pi_{k+1}^u)}{\nu_j^{i_l}}, \right.
$$

$$
\left. \min_{l \geq h+1} \min_{1 \leq j \leq m^l} \frac{c_j^{i_l} - J_j^{i_l}(\boldsymbol{\pi}_k) - \nu_j^{i_l} \sum_{u=1}^{h-1} D_{KL}^{max}(\pi_k^u, \pi_{k+1}^u)}{\nu_j^{i_l}} \right\}.
$$

*Note that $\delta^{i_1}$ (i.e., $h = 1$) is guaranteed to be non-negative if $\boldsymbol{\pi}_k$ satisfies safety constraints; that is because then $c_j^{i_l} \geq J_j^{i_l}(\boldsymbol{\pi}_k)$ for all $l$ and $j$, and the set $\{l \mid l < h\}$ is empty.*

*This formula for $\delta^{i_h}$, combined with Lemma 2, assures that the policies $\pi^{i_h}$ within $\delta^{i_h}$ max-KL distance from $\pi_k^{i_h}$ will not violate other agents' safety constraints, as long as the base joint policy $\boldsymbol{\pi}_k$ did not violate them (which assures $\delta^{i_1} \geq 0$). To see this, notice that for every $l = 1, \ldots, h-1$, and $j = 1, \ldots, m^l$,*

$$
D_{KL}^{max}(\pi_k^{i_h}, \pi^{i_h}) \leq \delta^{i_h} \leq \frac{c_j^{i_l} - J_j^{i_l}(\boldsymbol{\pi}_k) - L_{j,\boldsymbol{\pi}_k}^{i_l}(\pi_{k+1}^{i_l}) - \nu_j^{i_l} \sum_{u=1}^{h-1} D_{KL}^{max}(\pi_k^u, \pi_{k+1}^u)}{\nu_j^{i_l}},
$$

*implies* $J_j^{i_l}(\boldsymbol{\pi}_k) + L_{j,\boldsymbol{\pi}_k}^{i_l}(\pi_{k+1}^{i_l}) + \nu_j^{i_l} \sum_{u=1}^{h-1} D_{KL}^{max}(\pi_k^u, \pi_{k+1}^u) + \nu_j^{i_l} D_{KL}^{max}(\pi_k^{i_h}, \pi^{i_h}) \leq c_j^{i_l}.$

*By Lemma 2, the left-hand side of the above inequality is an upper bound of $J_j^{i_l}(\pi_{k+1}^{i_{1:h-1}}, \pi^{i_h}, \pi_k^{-i_{1:h}})$, which implies that the update of agent $i_h$ doesn't violate the constraint of $J_j^{i_l}$. The fact that the constraints of $J_j^{i_l}$ for $l \geq h + 1$ are not violated, i.e.,*

$$
J_j^{i_l}(\boldsymbol{\pi}_k) + \nu_j^{i_l} \sum_{u=1}^{h-1} D_{KL}^{max}(\pi_k^u, \pi_{k+1}^u) + \nu_j^{i_l} D_{KL}^{max}(\pi_k^{i_h}, \pi^{i_h}) \leq c_j^{i_l}, \tag{12}
$$

*is analogous.*

**Theorem 1.** *If a sequence of joint policies $(\boldsymbol{\pi}_k)_{k=0}^{\infty}$ is obtained from Algorithm 1, then it has the monotonic improvement property, $J(\boldsymbol{\pi}_{k+1}) \geq J(\boldsymbol{\pi}_k)$, as well as it satisfies the safety constraints, $J_j^i(\boldsymbol{\pi}_k) \leq c_j^i$, for all $k \in \mathbb{N}, i \in \mathcal{N}$, and $j \in \{1, \ldots, m^i\}$.*

*Proof.* Safety constraints are assured to be met by Remark 1. It suffices to show the monotonic improvement property. Notice that at every iteration $k$ of Algorithm 1, $\pi_k^{i_h} \in \overline{\Pi}^{i_h}$. Clearly $D_{KL}^{max}(\pi_k^{i_h}, \pi_k^{i_h}) = 0 \leq \delta^{i_h}$. Moreover,

$$
J_j^{i_h}(\boldsymbol{\pi}_k) + L_{j,\boldsymbol{\pi}_k}^{i_h}(\pi_k^{i_h}) + \nu_j^{i_h} D_{KL}^{max}(\pi_k^{i_h}, \pi_k^{i_h}) = J_j^{i_h}(\boldsymbol{\pi}_k) \leq c_j^{i_h} - \nu_j^{i_h} \sum_{l=1}^{h-1} D_{KL}^{max}(\pi_k^{i_l}, \pi_{k+1}^{i_l}),
$$

where the inequality is guaranteed by updates of previous agents, as described in Remark 1 (Inequality 12). By Theorem 1 from Schulman et al. (2015), we have

$$J(\pi_{k+1}) \geq J(\pi_k) + \mathbb{E}_{s \sim \rho_{\pi_k}, \mathbf{a} \sim \pi_{k+1}} \left[ A_{\pi_k}(s, \mathbf{a}) \right] - \nu D_{KL}^{\max}(\pi_k, \pi_{k+1}),$$

which by Equation 11 is lower-bounded by

$$\geq J(\pi_k) + \mathbb{E}_{s \sim \rho_{\pi_k}, \mathbf{a} \sim \pi_{k+1}} \left[ A_{\pi_k}(s, \mathbf{a}) \right] - \sum_{h=1}^{n} \nu D_{KL}^{\max}(\pi_k^{i_h}, \pi_{k+1}^{i_h})$$

which by Lemma 1 equals

$$= J(\pi_k) + \sum_{h=1}^{n} \mathbb{E}_{s \sim \rho_{\pi_k}, \mathbf{a}^{i_{1:h}} \sim \pi_{k+1}^{i_{1:h}}} \left[ A_{\pi_k}^{i_h}(s, \mathbf{a}^{i_{1:h-1}}, \mathbf{a}^{i_h}) \right] - \sum_{h=1}^{n} \nu D_{KL}^{\max}(\pi_k^{i_h}, \pi_{k+1}^{i_h})$$

$$= J(\pi_k) + \sum_{h=1}^{n} \left( L_{\pi_k}^{i_{1:h}}(\pi_{k+1}^{i_{1:h-1}}, \pi_{k+1}^{i_h}) - \nu D_{KL}^{\max}(\pi_k^{i_h}, \pi_{k+1}^{i_h}) \right), \tag{13}$$

and as for every $h$, $\pi_{k+1}^{i_h}$ is the argmax, this is lower-bounded by

$$\geq J(\pi_k) + \sum_{h=1}^{n} \left( L_{\pi_k}^{i_{1:h}}(\pi_{k+1}^{i_{1:h-1}}, \pi_k^{i_h}) - \nu D_{KL}^{\max}(\pi_k^{i_h}, \pi_k^{i_h}) \right),$$

which, as follows from Definition 1, equals

$$= \mathcal{J}(\pi_k) + \sum_{h=1}^{n} 0 = J(\pi_k), \text{ which finishes the proof.}$$

$\square$

## C AUXILIARY RESULTS FOR IMPLEMENTATION OF MACPO

**Theorem 2.** *The solution to the following problem*

$$p* = \min_{x} g^T x$$
$$s.t. \ b^T x + c \leq 0$$
$$x^T H x \leq \delta,$$

*where $g, b, x \in \mathbb{R}^n, c, \delta \in \mathbb{R}, \delta > 0, H \in \mathbb{S}^n$, and $H > 0$. When there is at least one strictly feasible point, the optimal point $x^*$ satisfies:*

$$x^* = -\frac{1}{\lambda_*} H^{-1} \left( g^T + v_* b \right)$$

*where $\lambda_*$ and $v_*$ are defined by*

$$v_* = \left( \frac{\lambda_* c - r}{s} \right)_+$$

$$\lambda_* = \arg\max_{\lambda \geq 0} \begin{cases} f_a(\lambda) \triangleq \frac{1}{2\lambda} \left( \frac{r^2}{s} - q \right) + \frac{\lambda}{2} \left( \frac{c^2}{s} - \delta \right) - \frac{rc}{s} & \text{if } \lambda c - r > 0 \\ f_b(\lambda) \triangleq -\frac{1}{2} \left( \frac{q}{\lambda} + \lambda \delta \right) & \text{otherwise} \end{cases}$$

*where $q = g^T H^{-1} g$, $r = g^T H^{-1} b$, and $s = b^T H^{-1} b$.*

*Furthermore, let $\Lambda_a \triangleq \{\lambda \mid \lambda c - r > 0, \lambda \geq 0\}$, and $\Lambda_b \triangleq \{\lambda \mid \lambda c - r \leq 0, \lambda \geq 0\}$. The value of $\lambda_*$ satisfies*

$$\lambda_* \in \left\{ \lambda_*^a \triangleq \text{Proj} \left( \sqrt{\frac{q - r^2/s}{\delta - c^2/s}}, \Lambda_a \right), \lambda_*^b \triangleq \text{Proj} \left( \sqrt{\frac{q}{\delta}}, \Lambda_b \right) \right\} \tag{14}$$

*where $\lambda_* = \lambda_*^a$ if $f_a \left( \lambda_*^a \right) > f_b \left( \lambda_*^b \right)$ and $\lambda_* = \lambda_*$ otherwise, and $\text{Proj}(a, S)$ is the projection of a point $x$ on to a set $S$. Note the projection of a point $x \in \mathbb{R}$ onto a convex segment of $\mathbb{R}$, $[a, b]$, has value $\text{Proj}(x, [a, b]) = \max(a, \min(b, x))$.*

*Proof.* See Achiam et al. (2017) (Appendix 10.2). $\qquad\qquad\square$

## D   MACPO

---

**Algorithm 2:** MACPO

---

1: **Input:** Stepsize $\alpha$, batch size $B$, number of: agents $n$, episodes $K$, steps per episode $T$, possible steps in line search $L$.

2: **Initialize:** Actor networks $\{\theta_0^i, \ \forall i \in \mathcal{N}\}$, Global V-value network $\{\phi_0\}$, Individual $V^i$-cost networks $\{\phi_{j,0}^i\}_{i=1:n, j=1:m}$, Replay buffer $\mathcal{B}$

3: **for** $k = 0, 1, \ldots, K - 1$ **do**

4:     Collect a set of trajectories by running the joint policy $\boldsymbol{\pi}_{\boldsymbol{\theta}_k} = (\pi_{\theta_k^1}^1, \ldots, \pi_{\theta_k^n}^n)$.

5:     Push transitions $\{(o_t^i, a_t^i, o_{t+1}^i, r_t), \forall i \in \mathcal{N}, t \in T\}$ into $\mathcal{B}$.

6:     Sample a random minibatch of $M$ transitions from $\mathcal{B}$.

7:     Compute advantage function $\hat{A}(s, \mathbf{a})$ based on global V-value network with GAE.

8:     Compute cost-advantage functions $\hat{A}_j^i(s, a^i)$ based on individual $V^i$-cost critics with GAE.

9:     Draw a random permutation of agents $i_{1:n}$.

10:     Set $M^{i_1}(s, \mathbf{a}) = \hat{A}(s, \mathbf{a})$.

11:     **for** agent $i_h = i_1, \ldots, i_n$ **do**

12:         Estimate the gradient of the agent's maximisation objective

$$\hat{\boldsymbol{g}}_k^{i_h} = \frac{1}{B} \sum_{b=1}^{B} \sum_{t=1}^{T} \nabla_{\theta_k^{i_h}} \log \pi_{\theta_k^{i_h}}^{i_h} \left( a_t^{i_h} \mid o_t^{i_h} \right) M^{i_{1:h}}(s_t, \boldsymbol{a}_t).$$

13:         **for** $j = 1, \ldots, m^{i_h}$ **do**

14:             Estimate the gradient of the agent's $j^{\text{th}}$ cost

$$\hat{\boldsymbol{b}}_j^{i_h} = \frac{1}{B} \sum_{b=1}^{B} \sum_{t=1}^{T} \nabla_{\theta_k^{i_h}} \log \pi_{\theta_k^{i_h}}^{i_h} \left( a_t^{i_h} \mid o_t^{i_h} \right) \hat{A}_j^{i_h}(s_t, a_t^{i_h}).$$

15:         **end for**

16:         Set $\hat{\boldsymbol{B}}^{i_h} = \left[ \hat{\boldsymbol{b}}_1^{i_h}, \ldots, \hat{\boldsymbol{b}}_m^{i_h} \right]$.

17:         Compute $\hat{\boldsymbol{H}}_k^{i_h}$, the Hessian of the average KL-divergence

$$\frac{1}{BT} \sum_{b=1}^{B} \sum_{t=1}^{T} D_{\text{KL}} \left( \pi_{\theta_k^{i_h}}^{i_h}(\cdot|o_t^{i_h}), \pi_{\theta^{i_h}}^{i_h}(\cdot|o_t^{i_h}) \right).$$

18:         Solve the dual (5) for $\lambda_*^{i_h}, \mathbf{v}_*^{i_h}$.
         Use the conjugate gradient algorithm to compute the update direction

$$\boldsymbol{x}_k^{i_h} = (\hat{\boldsymbol{H}}_k^{i_h})^{-1} \left( \boldsymbol{g}_k^{i_h} - \hat{\boldsymbol{B}}^{i_h} \mathbf{v}_*^{i_h} \right),$$

19:         Update agent $i_h$'s policy by

$$\theta_{k+1}^{i_h} = \theta_k^{i_h} + \frac{\alpha^j}{\lambda_*^{i_h}} \hat{\boldsymbol{x}}_k^{i_h},$$

         where $j \in \{0, 1, \ldots, L\}$ is the smallest such $j$ which improves the sample loss, and satisfies the sample constraints, found by the backtracking line search.

20:         **if** the approximate is not feasible **then**

21:             Use equation (6) to recover policy $\theta_{k+1}^{i_h}$ from unfeasible points.

22:         **end if**

23:         Compute $M^{i_{1:h+1}}(s, \mathbf{a}) = \dfrac{\pi_{\theta_{k+1}^{i_h}}^{i_h} \left( a^{i_h} | o^{i_h} \right)}{\pi_{\theta_k^{i_h}}^{i_h} \left( a^{i_h} | o^{i_h} \right)} M^{i_{1:h}}(s_t, \mathbf{a}_t)$. //Unless $h = n$.

24:     **end for**

25:     Update V-value network by following formula:

26:     $\phi_{k+1} = \arg \min_\phi \frac{1}{N} \frac{1}{T} \sum_{n=1}^{N} \sum_{t=0}^{T} \left( V_\phi(s_t) - \hat{R}_t \right)^2$

27: **end for**

---

# E  MAPPO-LAGRANGIAN

---

**Algorithm 3:** MAPPO-Lagrangian

---

1: **Input:** Stepsizes $\alpha_\theta, \alpha_\lambda$, batch size $B$, number of: agents $n$, episodes $K$, steps per episode $T$, discount factor $\gamma$.

2: **Initialize:** Actor networks $\{\theta_0^i, \ \forall i \in \mathcal{N}\}$, Global V-value network $\{\phi_0\}$,
   V-cost networks $\{\phi_{j,0}^i\}_{1 \le j \le m^i}^{i \in \mathcal{N}}$, Replay buffer $\mathcal{B}$.

3: **for** $k = 0, 1, \ldots, K-1$ **do**

4:    Collect a set of trajectories by running the joint policy $\boldsymbol{\pi}_{\theta_k} = (\pi_{\theta_k^1}^1, \ldots, \pi_{\theta_k^n}^n)$.

5:    Push transitions $\{(o_t^i, a_t^i, o_{t+1}^i, r_t), \forall i \in \mathcal{N}, t \in T\}$ into $\mathcal{B}$.

6:    Sample a random minibatch of $B$ transitions from $\mathcal{B}$.

7:    Compute advantage function $\hat{A}(\mathrm{s}, \mathbf{a})$ based on global V-value network with GAE.

8:    Compute cost advantage functions $\hat{A}_j^i(\mathrm{s}, \mathrm{a}^i)$ for all agents and costs,
      based on V-cost networks with GAE.

9:    Draw a random permutation of agents $i_{1:n}$.

10:   Set $M^{i_1}(\mathrm{s}, \mathbf{a}) = \hat{A}(\mathrm{s}, \mathbf{a})$.

11:   **for** agent $i_h = i_1, \ldots, i_n$ **do**

12:      Initialise a policy parameter $\theta^{i_h} = \theta_k^{i_h}$,
         and Lagrangian multipliers $\lambda_j^{i_h} = 0, \ \forall j = 1, \ldots, m^{i_h}$.

13:      Make the Lagrangian modification step of objective construction
$$M^{i_h,(\lambda)}(\mathrm{s}_t, \mathbf{a}_t) = M^{i_h}(\mathrm{s}_t, \mathbf{a}_t) - \sum_{j=1}^{n} \lambda_j^{i_h} \hat{A}_j^{i_h}(\mathrm{s}_t, \mathrm{a}_t^{i_h}).$$

14:      **for** $e = 1, \ldots, e_{\mathrm{PPO}}$ **do**

15:         Differentiate the Lagrangian PPO-Clip objective
$$\Delta_{\theta^{i_h}} =$$
$$\nabla_{\theta^{i_h}} \frac{1}{B} \sum_{b=1}^{B} \sum_{t=0}^{T} \min\left( \frac{\pi_{\theta^{i_h}}^{i_h}\left(a_t^{i_h} | o_t^{i_h}\right)}{\pi_{\theta_k^{i_h}}^{i_h}\left(a_t^{i_h} | o_t^{i_h}\right)} M^{i_h,(\lambda)}(\mathrm{s}_t, \boldsymbol{a}_t), \ \mathrm{clip}\left( \frac{\pi_{\theta^{i_h}}^{i_h}\left(a_t^{i_h} | o_t^{i_h}\right)}{\pi_{\theta_k^{i_h}}^{i_h}\left(a_t^{i_h} | o_t^{i_h}\right)}, 1 \pm \epsilon \right) M^{i_h,(\lambda)}(\mathrm{s}_t, \boldsymbol{a}_t) \right).$$

16:         Update temprorarily the actor paramaters
$$\theta^{i_h} \leftarrow \theta^{i_h} + \alpha_\theta \Delta_{\theta^{i_h}}.$$

17:         **for** $j = 1, \ldots, m^{i_h}$ **do**

18:            Approximate the constraint violation
$$d_j^{i_h} = \frac{1}{BT} \sum_{b=1}^{B} \sum_{t=1}^{T} \hat{V}_j^{i_h}(s_t) - c_j^{i_h}.$$

19:            Differentiate the constraint
$$\Delta\lambda_j^{i_h} = \frac{-1}{B} \sum_{b=1}^{B} \left( d_j^{i_h}(1 - \gamma) + \sum_{t=0}^{T} \frac{\pi_{\theta^{i_h}}^{i_h}(a_t^{i_h} | o_t^{i_h})}{\pi_{\theta_k^{i_h}}^{i_h}(a_t^{i_h} | o_t^{i_h})} \hat{A}_j^{i_h}(s_t, a_t^{i_h}) \right).$$

20:         **end for**

21:         **for** $j = 1, \ldots, m^{i_h}$ **do**

22:            Update temporarily the Lagrangian multiplier
$$\lambda_j^{i_h} \leftarrow \mathrm{ReLU}\left( \lambda_j^{i_h} - \alpha_\lambda \Delta\lambda_j^{i_h} \right).$$

23:         **end for**

24:      **end for**

25:      Update the actor parameter $\theta_{k+1}^{i_h} = \theta^{i_h}$.

26:      Compute $M^{i_{h+1}}(\mathrm{s}, \mathbf{a}) = \frac{\pi_{\theta_{k+1}^{i_h}}^{i_h}\left(\mathrm{a}^{i_h} | \mathrm{o}^{i_h}\right)}{\pi_{\theta_k^{i_h}}^{i_h}\left(\mathrm{a}^{i_h} | \mathrm{o}^{i_h}\right)} M^{i_h}(\mathrm{s}, \mathbf{a})$. //Unless $h = n$.

27:   **end for**

28:   Update V-value network (and V-cost networks analogously) by following formula:

29:   $\phi_{k+1} = \arg\min_\phi \frac{1}{BT} \sum_{b=1}^{B} \sum_{t=0}^{T} \left( V_\phi(s_t) - \hat{R}_t \right)^2$

30: **end for**

---

# F  SAFE MULTI-AGENT MUJOCO

Safe MAMuJoCo is an extension of MAMuJoCo (Peng et al., 2020). In particular, the background environment, agents, physics simulator, and the reward function are preserved. However, as oppose to its predecessor, Safe MAMuJoCo environments come with obstacles, like walls or bombs. Furthermore, with the increasing risk of an agent stumbling upon an obstacle, the environment emits cost (Brockman et al., 2016). According to the scheme from Zanger et al. (2021), we characterise the cost functions for each task below.

## MANYAGENT ANT & ANT

The width of the corridor set by two walls is $9m$(ManyAgent Ant), The width of the corridor set by three folding line walls with an angle of 30 degrees is $10m$(Ant). The environment emits the cost of 1 for an agent, if the distance between the robot and the wall is less than $1.8m$, or when the robot topples over. This can be described as

$$c_t = \begin{cases} 0, & \text{for} \quad 0.2 \le z_{\text{torso},t+1} \le 1.0 \text{ and } \left\| x_{\text{torso},t+1} - x_{\text{wall}} \right\|_2 \ge 1.8 \\ 1, & \text{otherwise} . \end{cases}$$

where $z_{\text{torso},t+1}$ is the robot's torso's $z$-coordinate, and $x_{\text{torso},t+1}$ is the robot's torso's $x$-coordinate, at time $t+1$; $x_{\text{wall}}$ is the $x$-coordinate of the wall.

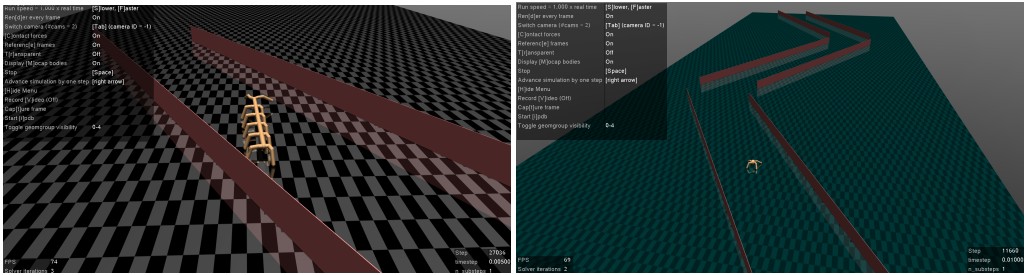

Figure 3: ManyAgent Ant 3x2 with a corridor and Ant 4x2 with three corridors.

## HALFCHEETAH & COUPLE HALFCHEETAH

In these tasks, the agents move inside a corridor (which constraints their movement, but does not induce costs). Together with them, there are bombs moving inside the corridor. If an agent finds itself too close to a bomb, the distance between an agent and a bomb is less than $9m$, a cost of 1 will be emitted.

$$c_t = \begin{cases} 0, & \text{for} \quad \left\| y_{\text{torso},t+1} - y_{\text{obstacle}} \right\|_2 \ge 9 \\ 1, & \text{otherwise} . \end{cases}$$

where $y_{\text{torso},t+1}$ is the $y$-coordinate of the robot's torso, and $y_{\text{obstacle}}$ is the $y$-coordinate of the moving obstacle.

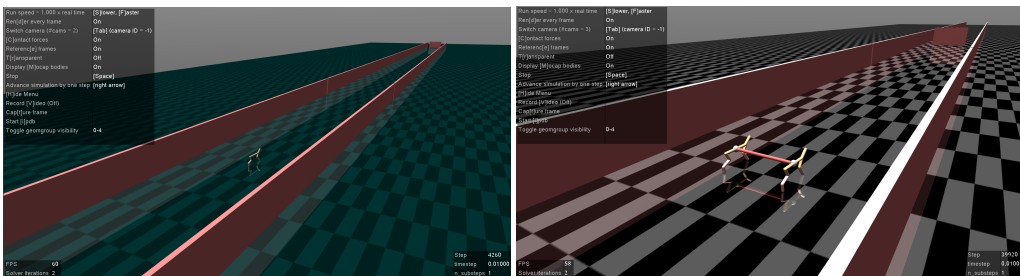

Figure 4: HalfCheetah 2x3 and Couple HalfCheetah 1P1.

# G    EXPERIMENTS IN SAFE MANY-AGENT ANT ENVIRONMENTS

We provide additional results on the Safe Many-Agent ant tasks.

The width of the corridor is $12m$; its walls fold at the angle of 30 degrees. The environment emits the cost of 1 for an agent, if the distance between the robot and the wall is less than $1.8m$, or when the robot topples over. This can be described as

$$c_t = \begin{cases} 0, & \text{for} \quad 0.2 \le z_{\text{torso},t+1} \le 1.0 \text{ and } \left\| x_{\text{torso},t+1} - x_{\text{wall}} \right\|_2 \ge 1.8 \\ 1, & \text{otherwise} . \end{cases}$$

where $z_{\text{torso},t+1}$ is the robot's torso's $z$-coordinate, and $x_{\text{torso},t+1}$ is the robot's torso's $x$-coordinate, at time $t + 1$; $x_{\text{wall}}$ is the $x$-coordinate of the wall.

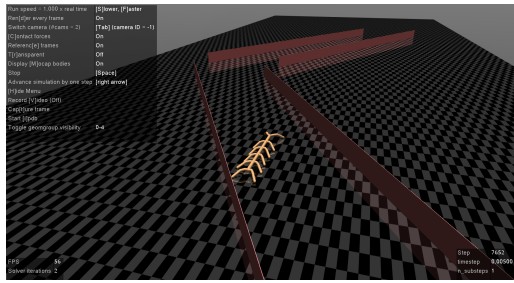

Figure 5: Many-Agent Ant 3x2 with two folding line walls.

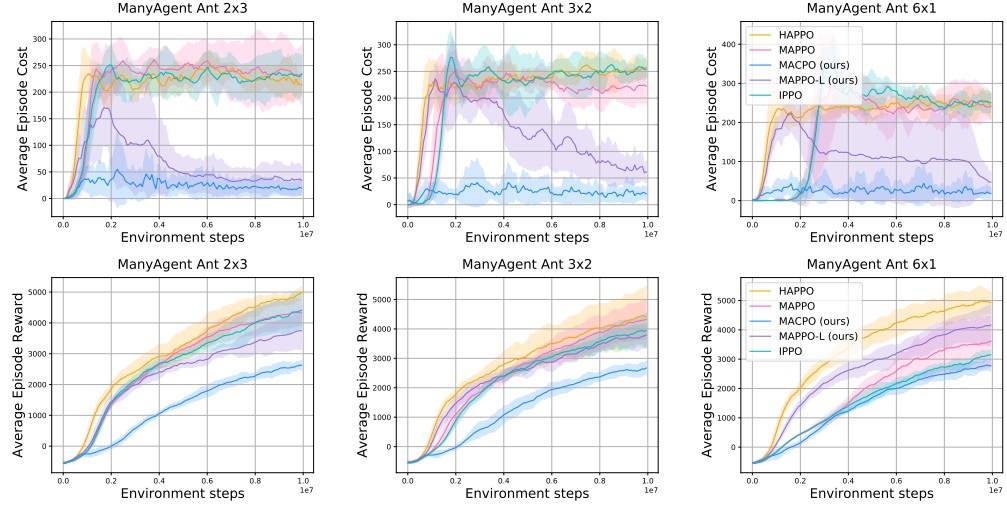

Figure 6: Performance comparisons on tasks of Safe ManyAgent Ant in terms of cost (the first row) and reward (the second row). The safety constraint values is set to 10. Our algorithms are the only ones that learn the safety constraints, while achieving satisfying performance in terms of the reward.

# H  DETAILS OF SETTINGS FOR EXPERIMENTS

In this section, we introduce the details of settings for our experiments. The code is available at https://github.com/Anonymous-ICLR2022/Multi-Agent-Constrained-Policy-Optimisation

| hyperparameters | value | hyperparameters | value | hyperparameters | value |
|---|---|---|---|---|---|
| critic lr | 5e-3 | optimizer | Adam | num mini-batch | 40 |
| gamma | 0.99 | optim eps | 1e-5 | batch size | 16000 |
| gain | 0.01 | hidden layer | 1 | training threads | 4 |
| std y coef | 0.5 | actor network | mlp | rollout threads | 16 |
| std x coef | 1 | eval episodes | 32 | episode length | 1000 |
| activation | ReLU | hidden layer dim | 64 | max grad norm | 10 |

Table 1: Common hyperparameters used for MAPPO-Lagrangian, MAPPO, HAPPO, IPPO, and MACPO in the Safe Multi-Agent MuJoCo domain

| Algorithms | MAPPO-Lagrangian | MAPPO | HAPPO | IPPO | MACPO |
|---|---|---|---|---|---|
| actor lr | 9e-5 | 9e-5 | 9e-5 | 9e-5 | / |
| ppo epoch | 5 | 5 | 5 | 5 | / |
| kl-threshold | / | / | / | / | 0.0065 |
| ppo-clip | 0.2 | 0.2 | 0.2 | 0.2 | / |
| Lagrangian coef | 0.78 | / | / | / | / |
| Lagrangian lr | 1e-3 | / | / | / | / |
| fraction | / | / | / | / | 0.5 |
| fraction coef | / | / | / | / | 0.27 |

Table 2: Different hyperparameters used for MAPPO-Lagrangian, MAPPO, HAPPO, IPPO, and MACPO in the Safe Multi-Agent MuJoCo domain.

| task | value | task | value | task | value |
|---|---|---|---|---|---|
| Ant(2x4) | 0.2 | Ant(4x2) | 0.2 | Ant(2x4d) | 0.2 |
| HalfCheetah(2x3) | 5 | HalfCheetah(3x2) | 5 | HalfCheetah(6x1) | 5 |
| ManyAgent Ant(2x3) | 1 | ManyAgent Ant(3x2) | 1 | ManyAgent Ant(6x1) | 1 |

Table 3: Safety bound used for MACPO in the Safe Multi-Agent MuJoCo domain

