# OpenReview forum: "Multi-Agent Constrained Policy Optimisation "
_ICLR.cc/2022/Conference — ICLR 2022 Submitted_

### Official Review · Reviewer_Aw6n · 2021-10-31

**Correctness:** 3
**Technical Novelty And Significance:** 3
**Empirical Novelty And Significance:** 2
**Recommendation:** 5
**Confidence:** 3

**Main Review:**

Strengths:
The algorithms are described well, and evaluations are provided. The problem is important.

Weaknesses:
1. Missing literature: For single agent, authors are suggested to see CRPO Xu et al. (2021), PDSC Chen et al. (2021), Triple-Q Wei et al. (2021), CSPDA Bai et al. (2021).
2. For multi-agent, see ECML paper https://2021.ecmlpkdd.org/wp-content/uploads/2021/07/sub_181.pdf which also gives a model-free approach. Comparisons between the approaches are needed.
3. There are no guarantees for the proposed approach. The combination of approaches does not seem to have enough novelty.
4. Although the constraints mentioned in Section 3 has the form of  "average constraints". I think what actually used in the experiments can be seen as a "peak constraint".  So I'm curious about  the performance of the algorithms  in plain "average constraints"(e.g. if the Cost function can return cost less than 0 when c_j=0). Moreover, since the constraint in experiments is "peak constraint", why not compare the two proposed algorithms to the simple baseline that add penalty toward the global reward once the constraint is violated.


**Summary Of The Paper:**

This paper proposes two algorithms for multi-agent reinforcement learning with constraints using policy optimization based approaches. The evaluations of the proposed approaches are provided.

**Summary Of The Review:**

It would be good to compare with the ECML paper approach mentioned above for scalability and performance.

---

> ### Author Response · Authors · 2021-11-18
> **Response to Reviewer Aw6n**
>
> ## We thank the reviewer Aw6n for their efforts in reviewing our paper, which will help us improve its quality.
>
> 1. > **Reviewer**: There are related works that should be mentioned. For example, CRPO Xu et al. (2021), PDSC Chen et al. (2021), Triple-Q Wei et al. (2021), and CSPDA Bai et. al. (2021).
>
> * **Response**: We understand relevancy and strengths of these works. However, as most of them were being developed concurrently, or released only recently, we did not have opportunity to come across them while working on this paper. We are happy to mention them in *Related Works* of our paper's final version. However, they also motivate us to highlight that what makes our methods stand out is their practicality, verified empirically. In the papers suggested by the reviewer, the proposed methods are either not evaluated at all, or are challenged against only a few simple tasks based on OpenAI gym. MACPO and MAPPO-Lagrangian, on the other hand, are shown to achieve near-SOTA performance in the challenging MAMuJoCo, while obeying safety constraints.
>
> 2. > **Reviewer**: The authors should compare their methods to CMIX.
>
> * **Response**: The paper that proposes CMIX *[Liu et al., 2021]* does not provide a link to the code.
>
> 3. > **Reviewer**: The proposed appraoch does not have theoretical guarantees. It combines two approaches, which is not novel enough.
>
> * **Response**: Knowing our domain, we can confidently say that MACPO and MAPPO-Lagrangian exceptionally-well theoretically justified. Indeed, its theory leverages results from CPO *[Achiam et al., 2017]* and HATRPO *[Kuba et al., 2021]*, but this enables derivation of methods which guarantee safety and reward optimisation (see our Theorem 1). Of course, our implementation deviates slightly from theory, but this is an inherent sacrifice one must make to train neural network policies: in *[Silver et al., 2014]*, the authors remark that even policy gradient estimators for neural networks are biased.
> We were quite surprised by this comment, given the reviewer's point 2. The method mentioned, CMIX, is an extension of QMIX *[Rashid et al., 2018]* through modification of the network architecture (the novelty aspect). Furthermore, it builds upon QMIX's monotonic value function factorisation, which does not hold in general, thus leaving the method without theoretical guarantees.
>
> 4. > **Reviewer**: The constraint framework used in the theoretical part of the paper is *average cost*, while in the experiments the authors use *peak cost*. It would be interesting to show the performance of the methods in the average cost setting.
>
> * **Response**: We used the **average cost** framework in the experiments (see the code: macpo/algorithms/r_mappo/r_macpo.py -> line 342).
> If the confusion was caused by the fact that the cost achieved was very low, while maintaining high reward performance, then we would like to highlight that this is the strength of our algorithms. They can learn powerful policies by obeying very strict safety constraints.
>
> ***Reference***
>
> -[Silver et al., 2014] Silver, David, et al. "Deterministic policy gradient algorithms." International conference on machine learning. PMLR, 2014.
>
> -[Achiam et al., 2017] Achiam, Joshua, et al. "Constrained policy optimization." International Conference on Machine Learning. PMLR, 2017.
>
> -[Rashid et al., 2018] Rashid, Tabish, et al. "Qmix: Monotonic value function factorisation for deep multi-agent reinforcement learning." International Conference on Machine Learning. PMLR, 2018.
>
> -[Kuba et al., 2021] Kuba, Jakub Grudzien, et al. "Trust region policy optimisation in multi-agent reinforcement learning." arXiv preprint arXiv:2109.11251 (2021).
>
> -[Liu et al., 2021] Liu, Chenyi, et al. "CMIX: Deep Multi-agent Reinforcement Learning with Peak and Average Constraints." Joint European Conference on Machine Learning and Knowledge Discovery in Databases. Springer, Cham, 2021.

---

### Official Review · Reviewer_rBTj · 2021-11-03

**Correctness:** 3
**Technical Novelty And Significance:** 3
**Empirical Novelty And Significance:** 3
**Recommendation:** 5
**Confidence:** 3

**Main Review:**

**Strengths**

1. This paper addressed an interesting and important question in MARL, i.e., how to learn the optimal policy with certain constraints.

2. Their method was shown to be valid and promising both theoretically and empirically.

**Weaknesses**

1. The main idea of this work is to combine and extend [1] and [2], while there is no sufficient discussion on the methodological and theoretical contribution of this work beyond these two cited works. More necessary elaboration and justification should be made to solid the novelty of this paper.

[1] Joshua Achiam, David Held, Aviv Tamar, and Pieter Abbeel. Constrained policy optimization. In
International Conference on Machine Learning, pp. 22–31. PMLR, 2017.

[2] Jakub Grudzien Kuba, Ruiqing Chen, Munning Wen, Ying Wen, Fanglei Sun, Jun Wang, and Yaodong Yang. Trust region policy optimisation in multi-agent reinforcement learning. arXiv preprint arXiv:2109.11251, 2021a.

2. This paper particularly cares about a safe constraint, while the relationship between cost and reward is not very clear. Based on MAPPO-Lagrangian where the authors use a hard constraint on satisfying the safe condition, it seems that the cost itself is more important than reward. However, in the real application, there may exist a trade-off between cost and reward. I felt some human preference or domain knowledge as required for the constraint optimization literature.



**Summary Of The Paper:**

This paper considers the multi-agent reinforcement learning (MARL) problem with safety constraints. The authors proposed two methods, Multi-Agent Constrained Policy Optimisation (MACPO) and MAPPO-Lagrangian, by leveraging the theories from both constrained policy optimization and multi-agent trust-region learning. Their method is shown valid both theoretically and empirically. My main concerns lie in their novelty compared with existing literature and the problem set.

**Summary Of The Review:**

I think this is a borderline paper that addressed an important question with reasonably good performance while lacking necessary elaboration and justification on their novelty, as commented in my 'Main Review'. I am willing to upgrade if my concerns can be addressed during the rebuttal period.

---

> ### Author Response · Authors · 2021-11-18
> **Response to Reviewer rBTj**
>
> 1. > **Reviewer**: To solve the important safety problem in MARL, this work combines results from CPO *[Achiam et al., 2017]* and HATRPO *[Kuba et al., 2021]*. A discussion on the novelty of the provided solutions is missing.
>
> * **Response**: We agree that these two works inspired development of our techniques. However, in his/her opinion on our novelty, the reviewer may have missed the fact that the problem of safety in MARL itself has not been thoroughly studied. While we have discussed its importance with regard of the practical deployment of A.I., very few works considered modeling this problem mathematically. Our model (which we also justified) is one of the first such frameworks.
> As for the novelty of our techniques, we agree that we have not provided extensive description of it. Here we present some suggestions. (1) Our algorithms enable agents to monotonically improve their policies, while meeting their own and **other agents'** safety constraints. Although reward and local-cost improvements seem to be a combination of CPO and HATRPO, the satisfaction of other agents' constraints follows from Remark 1 (Appendix B)---a sufficiently small KL-constraint controls the change of other agents' cost. In fact, we proved (and can add to the paper's final version) that the gradient, with respect to agent $i$, of agent $j$'s cost is zero. (2) *MAPPO-Lagrangian* enables one to perform very heavily-constrained optimisation very efficiently, with only first-order differentiation---the algorithm optimises the return, while meeting the policy-divergence and (possibly multiple) cost constraints. This, to our knowledge, is the first such an efficient safe algorithm in general RL at all. Even in the single-agent case, PPO-Lagrangian was reported to perform poorly with respect to return *[Ray et al., 2019]*.
> Lastly, we highlight the development of *Safe Multi-Agent MuJoCo*. One of the reasons why Safe-MARL has not been intensely tackled is a lack of a suitable benchmark; a hole which we have filled.
>
> 2. > **Reviewer**: The paper is particularly focused on the cost minimisation, while there may be an underlying relationship between the cost and the reward. Improsing inequality constraints on the expected cost seems to require human knowledge.
>
> * **Response**: Indeed, from the moment we formulated the safe-MARL problem as a constrained optimisation, we do not consider an optimal trade-off between rewards and costs. However, we do not focus on the cost minimisation. The methods are developed to keep the cost under certain thresholds, not to minimise them to zero. Furthermore, when it comes to the constrained formulation, we think that this is the essence of safety considerations: we set hard limits against unfortunate/undesirable events. For a example, for a self-driving car, we want to keep the number of pedestrians hurt zero. Indeed, this requires some domain knowledge, but so does implementation of the reward function (reward a car for miles covered), and the state (field of view). The constrained formulation has been a standard in RL *[Achiam et al., 2017, Ray et al., 2019]*.
>
> ***Reference***
>
> -[Schulman et al., 2017] Schulman, John, et al. "Proximal policy optimization algorithms." arXiv preprint arXiv:1707.06347 (2017).
> -[Achiam et al., 2017] Achiam, Joshua, et al. "Constrained policy optimization." International Conference on Machine Learning. PMLR, 2017.
>
> -[Ray et al., 2019] Ray, Alex, Joshua Achiam, and Dario Amodei. "Benchmarking safe exploration in deep reinforcement learning." arXiv preprint arXiv:1910.01708 7 (2019).
>
> -[Kuba et al., 2021] Kuba, Jakub Grudzien, et al. "Trust region policy optimisation in multi-agent reinforcement learning." arXiv preprint arXiv:2109.11251 (2021).

---

### Official Review · Reviewer_MyQj · 2021-11-03

**Correctness:** 3
**Technical Novelty And Significance:** 2
**Empirical Novelty And Significance:** 2
**Recommendation:** 5
**Confidence:** 3

**Main Review:**

Strength:

This paper proposes two interesting algorithms to solve the safe RL problem in the multi-agent setting based on CPO, PPO and primal-dual approches. The author provides theoretical guarantee for their proposed algorithm (Thm 1). The effectiveness of their proposed algorithms is supported by extensive experiments.

Weakness:

1. More discussions are need to highlight the novelty and challenging compare with CPO (Achiam 2017).

2. The theoretical result (Thm1) can be improved if the author can provide quantitative characterization of policy improvement and constraint satisfication similar to Proposition 1 and Proposition 2 in CPO (Achiam 2017), respectively.

2. The experiment part misses some details, which I list below:

    (1) It is not clear how to apply the algorithm MACPO in the multi-constrains setting in experiment since solving eq. (4) in the multi- constraints setting is very challenging.

    (2) It is also not clear how MACPO updates the policy at the beginning of the algorithm, when the initialization point is infeasible. From the theoretical results, it seems that the constraint satisfications can only be guaranteed after the algorithm enter into the feasible region.

2. The update rule of MACPO and MAPPO-LAGRANGIAN seems to be not efficient when the algorithm is not feasible. Because in the infeasible region we hope the algorithm can enter into the feasible region as soon as possible. Therefore, instead of optimizing a mixed objective function of both reward and cost, we may want to only minizing the cost.

**Summary Of The Paper:**

This paper studies the safe RL in the multi-agent setting. Specifically, the author leverage the theories from constrained policy optimization and multi-agent trust region learning to propose two algorithms: MACPO and MAPPO-Lagrangian. From the theoretical side, the author shows that in the idea setting, the proposed algorithm is guarantee to improve the objective function at iteration and the constraint satisfications can always be guaranteed. The author also demonstrate the effectiveness of their proposed algorithms in the new environment named "Safe MAMuJoCo".

**Summary Of The Review:**

I can consider raise my score if the author can address my question in the "main review".

---

> ### Author Response · Authors · 2021-11-18
> **Response to Reviewer MyQj**
>
> ## We thank the reviewer MyQj for their efforts in reviewing our paper, which will help us improve its quality.
>
> 1. > **Reviewer**: Algorithm 1 assumes that the policies the agents start with are feasible, which is not realistic. How do they update their policies if the initial policy is unsafe? Equation (6) seems to be a slow method for that.
>
> * **Response**: Indeed, for convenience, we assumed feasiblity of the initial policy. As the reviewer mentioned, we arrive at the feasible set with Equation (6)---hence, in practice, the first passage of iterations always perform this update. As for this method's aim and practicality: Equation (6) indeed **does** ignore the optimisation objective, and focuses entirely on the crucial---safety. It minimises the expected cost with a TRPO-like update; as the cost inherits algebraic properties of the reward, this approach guarantees that the cost is monotonically (and rapidly) decreasing *[Schulman et al., 2015]*. This is reflected in empirical results on Figure 2.
>
> 2. > **Reviewer**: Is it possible to apply MACPO to a setting with multiple constraints?
>
> * **Response**: Yes, it is possible. One can implement it with a loop of primal-dual optimisation. In our case, it takes a similar form to the one from CPO *[Achiam et al., 2017]* (see Appendix 10.3.3). Indeed, however, with the growing number of constraints, this approach becomes less practical. That is exactly the reason why we also propose MAPPO-Lagrangian. Perhaps, we did not highlight it sufficiently in Section 4.3, but this new method differs from (already efficient) HAPPO in complexity by computing and differentiating $m^{i_h}$ constants for each agent $i_h$ (see Appendix E, Algorithm 3, lines 17-19). It is an algorithm that guarantees safety and performance in the case of multiple constraints.
>
> ***Reference***
> -[Achiam et al., 2017] Achiam, Joshua, et al. "Constrained policy optimization." International Conference on Machine Learning. PMLR, 2017.
>
> -[Schulman et al., 2017] Schulman, John, et al. "Proximal policy optimization algorithms." arXiv preprint arXiv:1707.06347 (2017).

---

### Decision · Program_Chairs · 2022-01-20

**Decision:**

Reject

**Comment:**

The paper addresses safe multi-agent reinforcement learning and makes two key contributions. First is a safety concerned multi agent benchmark, which is an extension of MAMuJoCo. Second, is the formulation and two solution to safety MARL problem. The authors pose safe MARL, and MARL problem with safety constraints, as a constrained Markov game.

The safety constrained MARL is an important, difficult, and understudied problem. The problem is more difficult that the single agent safe RL because of the non-stationarity in the MARL setting, which renders any theoretical guaranties conditioned on the assumptions of the behaviors of other agents. The authors are right to point out the lack of the benchmarks in the space.

That said, reflecting on the reviewers' feedback and my own reading of the paper, this paper is attempting to do too much (benchmark, problem formulation, and two methods), in too little space, and is falling short. For example, the benchmark is an important contribution, but it is barely mentioned in the main text of the paper. If this was fully safety benchmark paper, there is an opportunity to go beyond MAMuJoCo, which feels like a forced multi-agent problem, and construct a safety benchmark with energy constraints, cooperative and competitive tasks etc... If this was fully methods paper, there would be an opportunity for more in-depth analysis of the results that the reviewers' pointed out. In it's current form, the paper feels like proposing a benchmark not grounded in a real world problem, and then a method to solve the problem.

I would suggest the authors to either:
- submit the paper to a journal where a space constraint would not be in a way, or
- split it into two papers, a more comprehensive benchmark, and methods paper evaluated on more difficult problems.

Minor:
- Please update the literature. Some of the papers have been published, and they are cited as Arxiv papers.